# Indoor Visible-Light 3D Positioning System Based on GRU Neural Network

**Wuju Yang** **, Ling Qin *** **, Xiaoli Hu and Desheng Zhao**

College of Information Engineering, Inner Mongolia University of Science and Technology, Baotou 014010, China; yangwuju@stu.imust.edu.cn (W.Y.); huxiaoli@imust.edu.cn (X.H.); 2018978@imust.edu.cn (D.Z.)
* Correspondence: qinling1979@imust.edu.cn

**Abstract:** With the continuous development of artificial intelligence technology, visible-light positioning (VLP) based on machine learning and deep learning algorithms has become a research hotspot for indoor positioning technology. To improve the accuracy of robot positioning, we established a three-dimensional (3D) positioning system of visible-light consisting of two LED lights and three photodetectors. In this system, three photodetectors are located on the robot's head. We considered the impact of line-of-sight (LOS) and non-line-of-sight (NLOS) links on the received signals and used gated recurrent unit (GRU) neural networks to deal with nonlinearity in the system. To address the problem of poor stability during GRU network training, we used a learning rate attenuation strategy to improve the performance of the GRU network. The simulation results showed that the average positioning error of the system was 2.69 cm in a space of 4 m $\times$ 4 m $\times$ 3 m when only LOS links were considered and 2.66 cm when both LOS and NLOS links were considered with 95% of the positioning errors within 7.88 cm. For two-dimensional (2D) positioning with a fixed positioning height, 80% of the positioning error was within 9.87 cm. This showed that the system had a high anti-interference ability, could achieve centimeter-level positioning accuracy, and met the requirements of robot indoor positioning.

**Keywords:** robot; visible-light positioning (VLP); three-dimensional (3D); line-of-sight (LOS) and non-line-of-sight (NLOS) links; gated recurrent units (GRU) neural networks; learning rate decay strategy



## 1. Introduction

With the progress of human beings and the development of technology, the application scenarios of robots have become more complex and diversified, and robots need to complete more difficult and intelligent work. In order to improve the efficiency and performance of robots, the positioning and navigation of autonomous robots are essential. At present, wireless positioning technologies such as wireless local area networks (WLANs), Bluetooth, radio frequency identification (RFID), ZigBee, and ultra-wideband (UWB) are commonly used for indoor positioning [1–5], but these wireless technologies generally have disadvantages such as high electromagnetic radiation, high deployment costs, and low positioning accuracy [6]. Compared with these wireless technologies, visible-light has the advantages of abundant bandwidth resources, no electromagnetic pollution, and low equipment costs, and it can achieve lighting and positioning at the same time. As a new type of wireless positioning technology, visible-light positioning based on LED has become a research hotspot in the field of wireless positioning [7].

In recent years, with the development of artificial intelligence, machine learning and deep learning algorithms, with their strong self-learning and generalization abilities, have become able to provide accurate positioning results in the context of VLP, and increasing numbers of people have applied them to indoor visible-light positioning. Abu Bakar et al. [8] use a weighted k-nearest neighbor (WKNN) algorithm for localization in a fingerprint recognition technique based on received signal strength (RSS). The results show

that the positioning accuracy of the WKNN algorithm is better than that of the multi-layer perceptron (MLP)-based regressor. In addition to using a single ML algorithm, multiple ML algorithms can also be used for fusion localization. Huy Q. Tran et al. [9] use a dual-functional ML algorithm leveraging machine learning classification (MLC) and machine learning regression (MLR) functions to improve localization accuracy under the negative effects of multipath reflection. They use ML classification functions to divide the floor of a room into two separate zones. Then, the regression function of the ML algorithm is used to predict the position of the optical receiver. ML algorithms can also analyze and optimize other parameters. Sheng Zhang et al. [10] use neural networks to reduce position offset errors caused by uneven initial delay patterns of off-the-shelf LEDs. However, applying ML algorithms in VLP also has limitations, as they often require propagation of near-ideal behavior of the model and its parameters to perform well, and ML algorithms are too data-dependent and require a lot of time to be measured offline. We can obtain the data set by linear fitting, which can effectively reduce the offline measurement time. In addition, the parameter settings in the ML algorithm have a great influence on the positioning results, and the best model is not obtained frequently. Therefore, we need to call parameters or process them using optimization algorithms.

At present, most work on indoor visible-light localization has focused on two-dimensional positioning, assuming a fixed receiver height and ignoring positional errors due to height variation [11–13]. In the future, robots will need to complete a variety of difficult actions, so their positioning height cannot be limited, and they will require accurate and reliable three-dimensional positioning covering indoor areas. However, some 3D visible-light positioning systems use hybrid algorithms [14–16], which greatly increase the complexity of the system. In order to improve the accuracy of robot positioning and reduce system complexity, we proposed a three-dimensional indoor visible-light localization system based on a GRU neural network. We use three PDs as receivers and two LED light sources as transmitters, each LED sends signals of different frequencies. The signals collected by PD are filtered to obtain two signals of different frequencies. When data is processed, it is usually processed sequentially, so the collected data can be considered a kind of sequential data. Recurrent neural networks are very efficient for data with sequential properties, and can mine time series information from the data. The GRU network is a variant of the recurrent network, which can automatically extract effective features from experimental data, so as to obtain high positioning performance, and the localization model structure is simple and converges easily. In this study, considering the influence of LOS and NLOS links on the received signal strength, the GRU algorithm was applied to a three-dimensional indoor visible-light positioning system; a fingerprint database was established using the optical power value and position data received by the PD and then substituted into the GRU neural network to train the model; and, finally, the position information was predicted by the trained model, and the feasibility of the proposed algorithm was proved by simulations. As far as we know, the traditional 3D VLP positioning method requires the use of three or more LEDs to accurately position and does not consider wall reflections [17,18]. In addition, they usually use multiple localization algorithms in the selection of positioning algorithms, which makes the VLP system model complex. Compared to these articles, our proposed VLP system uses only two LEDs and a new receiver model, which is lower cost and easier to implement. We only use one positioning algorithm to achieve accurate three-dimensional positioning, and the system complexity is low. We analyze the influence of positioning height on the VLP system.

The rest of this paper is organized as follows: Section 2 describes the composition of the visible-light localization system model. Section 3 describes the principles of the GRU neural network. Then, in Section 4, the application of the GRU neural network for visible-light localization is described. Finally, the positioning results are discussed in Section 5, and the performance of the visible-light positioning system is analyzed.

## 2. Visible-Light Positioning Model

### 2.1. System Model

The indoor visible-light localization model designed in this study is shown in Figure 1. The room size was set to 4 m × 4 m × 3 m, and the corner of the room was used as the origin to establish the Cartesian coordinate system of the space. We used two LEDs as transmitters, placed on the ceiling, and each LED sent signals of different frequencies.

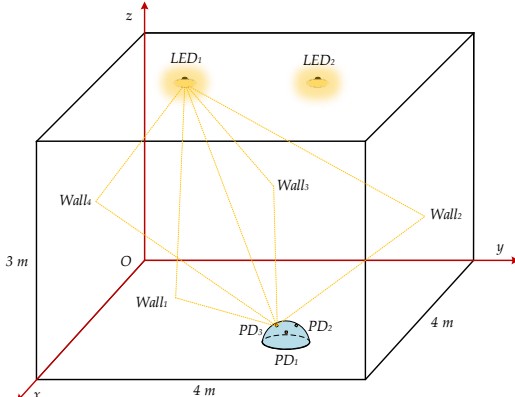

**Figure 1.** Indoor visible-light positioning model.

To fully receive the signal sent by the transmitter, we used three PDs as receivers, which were located in front of the robot's head, on the left at the rear, and on the right at the rear. The model structure of the robot head receiver is shown in Figure 2, which represents the robot head as a hemispherical model, and the three PDs on the head and the top center point are equidistant. In this robot head receiver model, the top center point $O$ was used as the test point; $r$ is the radius of the hemisphere; $l$ is the length of the arc between point $O$ and $PD_i$; $\alpha_i (i = 1, 2, 3)$ is the azimuth angle of $PD_i$; $\theta$ is the central angle of the arc between point $O$ and $PD_i$; and $\beta(0 < \beta < 90°)$ is the elevation angle of $PD_i$, which can be expressed as the following.

$$\beta = \theta = l/r \tag{1}$$

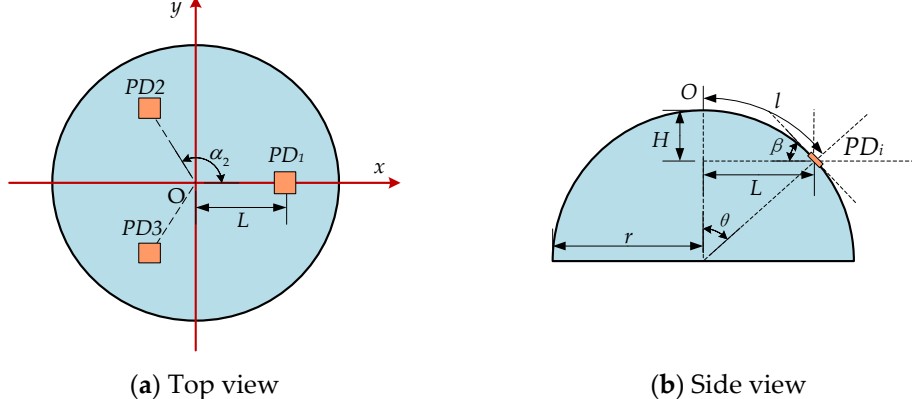

(**a**) Top view                                    (**b**) Side view

**Figure 2.** Robot head receiver model structure.

Therefore, the relationship between the position $(x_i, y_i, z_i)$ of $PD_i$ and the position $(x_0, y_0, z_0)$ of the top center point $O$ is

$$\begin{cases} x_i = x_0 + L\cos(\alpha_i) \\ y_i = y_0 + L\sin(\alpha_i) \\ z_i = z_0 - H \end{cases}, \tag{2}$$

where $L$ is the horizontal distance between point $O$ and $PD_i$, and $H$ is the vertical distance between point $O$ and $PD_i$. $L$ and $H$ can be expressed as the following.

$$L = r\sin(\beta), \tag{3}$$

$$H = r(1 - \cos(\beta)). \tag{4}$$

*2.2. Channel Model*

The indoor visible-light channel model is shown in Figure 3 for the direct link model and the reflected link model, respectively. For an LOS link model, the indoor optical signal transmission link is short, so the attenuation of the optical signal caused by absorption and scattering is small. However, for an NLOS link model, because the indoor walls, floors, and other objects with reflection characteristics cause the diffuse reflection of the optical signal, the optical signal transmission link becomes longer, increasing the attenuation of the optical signal. Therefore, we considered the transmission of optical signals through LOS and NLOS links. This not only conformed to the real-world environment but also allowed further study of the adverse effects of reflection on system performance, making the positioning system more reliable and practical.

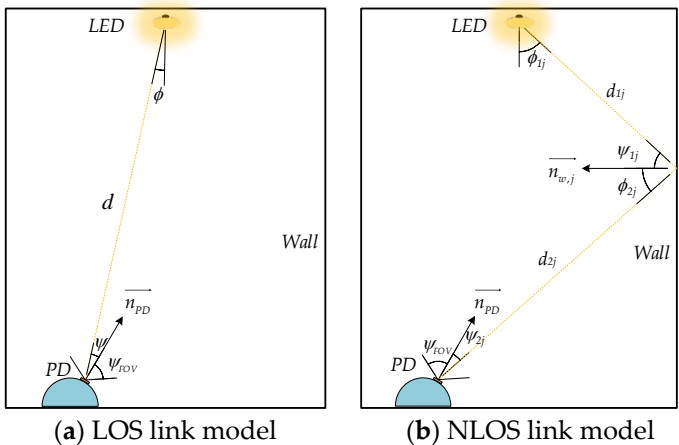

(**a**) LOS link model      (**b**) NLOS link model

**Figure 3.** Indoor visible-light channel model.

In the LOS link model, the relationship between the received power $P_{LOS}$ of the PD and the LED transmitted power $P_t$ can be expressed as [19].

$$P_{LOS} = P_t H_{LOS}(0), \tag{5}$$

where $H_{LOS}(0)$ is the DC gain of the LOS link. Assuming that the LEDs obey the Lambert radiation model, $H_{LOS}(0)$ can be expressed as [20]

$$H_{LOS}(0) = \begin{cases} \frac{(m+1)A_{PD}}{2\pi d^2}\cos^m(\phi)T_s(\psi)g(\psi)\cos(\psi), & 0 \leq \psi \leq \psi_{FOV} \\ 0, & \psi > \psi_{FOV} \end{cases}, \tag{6}$$

where $A_{PD}$ is the effective receiving area of the PD; $d$ is the distance from the PD to the LED; $m$ is the Lambertian emission order; $\phi$ is the emission angle of the LED; $T_s(\psi)$ is the optical filter gain; $g(\psi)$ is the gain of the optical concentrator; and $\psi$ and $\psi_{FOV}$ are the incidence and field-of-view (FOV) angles of the PD, respectively. $m$ and $g(\psi)$ can be expressed as [21]

$$m = -\frac{\ln(2)}{\ln(\cos(\phi_{1/2}))}, \tag{7}$$

$$g(\psi) = \begin{cases} \dfrac{n^2}{\sin^2(\psi_{FOV})}, & 0 \leq \psi \leq \psi_{FOV} \\ 0, & \psi > \psi_{FOV} \end{cases}, \tag{8}$$

where $\phi_{1/2}$ is the semi-angle at half-power of the LED emitters, and $n$ is the internal refractive index of the optical concentrator. In this paper, two LEDs placed on the ceiling were used as light sources, and three PDs placed on the hemispherical surface were used as receivers. Each PD had a certain inclination angle, and the radiating angle cosine of the LED and the incidence angle cosine of the inclined PD could be expressed as [22]

$$\cos(\phi) = \frac{h}{d}, \tag{9}$$

$$\cos(\psi) = \frac{\overrightarrow{v_{PD\_LED}} \cdot \overrightarrow{n_{PD}}}{\left\| \overrightarrow{v_{PD\_LED}} \right\| \left\| \overrightarrow{n_{PD}} \right\|}, \tag{10}$$

where $h$ is the vertical height of the LED in relation to the PD; $\overrightarrow{v_{PD\_LED}}$ is the direction vector from the PD to the LED; and $\overrightarrow{n_{PD}}$ is the normal vector of the PD receiving surface, which can be expressed as

$$\overrightarrow{n_{PD}} = (\cos(\alpha_r)\sin(\beta_r), \sin(\alpha_r)\sin(\beta_r), \cos(\beta_r)), \tag{11}$$

where $\alpha_r$ and $\beta_r$ are the azimuth and tilt angles of the PD, respectively. If the LED position coordinates were $(x_t, y_t, z_t)$, and the PD position coordinates were $(x_r, y_r, z_r)$, then from Equations (10) and (11) we could obtain the incidence angle cosine of the inclined PD to receive LED light as follows:

$$\cos(\psi) = \frac{(x_t - x_r)\cos(\alpha_r)\sin(\beta_r) + (y_t - y_r)\sin(\alpha_r)\sin(\beta_r) + (z_t - z_r)\cos(\beta_r)}{\sqrt{(x_t - x_r)^2 + (y_t - y_r)^2 + (z_t - z_r)^2}}. \tag{12}$$

In a primary reflective NLOS link, the relationship between the received power $P_{NLOS}$ of the PD and the LED transmitted power $P_t$ can be expressed as

$$P_{NLOS} = P_t H_{NLOS}(0), \tag{13}$$

where $H_{NLOS}(0)$ is the DC gain of the primary reflected NLOS link, which can be expressed as [23]

$$H_{\text{NLOS}}(0) = \begin{cases} \sum\limits_{j}^{N} \dfrac{A_{\text{PD}}(m+1)\rho\Delta A}{2\pi^2 d_{1j}^2 d_{2j}^2} \cos^m(\phi_{1j})\cos(\psi_{1j})\cos(\phi_{2j})\cos(\psi_{2j})T_s(\psi_{2j})g(\psi_{2j}), & 0 \leq \psi_{2j} \leq \psi_{FOV} \\ 0, & \psi_{2j} > \psi_{FOV} \end{cases}, \tag{14}$$

where $N$ indicates the number of all reflective walls divided by $\Delta A$ as the area element; $\rho$ is the reflectivity of the wall; $d_{1j}$ is the distance between the LED and the wall reflective element; $d_{2j}$ is the distance between the wall reflective element and the PD; $\phi_{1j}$ is the LED emission angle; $\psi_{1j}$ and $\phi_{2j}$ are the incidence and emission angles of the wall reflective element, respectively; and $\psi_{2j}$ is the incidence angle of the PD. If the normal vector $\overrightarrow{n_{w,j}}$ of the wall reflecting element is

$$\overrightarrow{n_{w,j}} = (\cos(\alpha_{w,j})\sin(\beta_{w,j}), \sin(\alpha_{w,j})\sin(\beta_{w,j}), \cos(\beta_{w,j})), \tag{15}$$

where $\alpha_{w,j}$ and $\beta_{w,j}$ are the azimuth and tilt angles of the wall reflector element, respectively, then the cosine corresponding to $\phi_{1j}, \psi_{1j}, \phi_{2j}$, and $\psi_{2j}$ can be expressed as

$$\cos(\phi_{1j}) = \frac{h_{1j}}{d_{1j}}, \tag{16}$$

$$\cos(\psi_{1j}) = \frac{(x_t - x_{w,j})\cos(\alpha_{w,j})\sin(\beta_{w,j}) + (y_t - y_{w,j})\sin(\alpha_{w,j})\sin(\beta_{w,j}) + (z_t - z_{w,j})\cos(\beta_{w,j})}{d_{1j}}, \tag{17}$$

$$\cos(\phi_{2j}) = \frac{(x_r - x_{w,j})\cos(\alpha_{w,j})\sin(\beta_{w,j}) + (y_r - y_{w,j})\sin(\alpha_{w,j})\sin(\beta_{w,j}) + (z_r - z_{w,j})\cos(\beta_{w,j})}{d_{2j}}, \tag{18}$$

$$\cos(\psi_{2j}) = \frac{(x_{w,j} - x_r)\cos(\alpha_r)\sin(\beta_r) + (y_{w,j} - y_r)\sin(\alpha_r)\sin(\beta_r) + (z_{w,j} - z_r)\cos(\beta_r)}{d_{2j}}, \tag{19}$$

where $h_{1j}$ is the vertical height of the LED in relation to the wall reflector element, and $(x_{w,j}, y_{w,j}, z_{w,j})$ are the position coordinates of the wall reflector element.

In the VLP system, each LED is installed in a vertical ceiling downward fashion, with its half-power half-angle set to $30°$, which means the amount of light that the ceiling receives directly from the LED bulb is limited. We design the robot's shell with a low-reflectivity material, so we do not take into account the reflection of the robot itself. The receiver is mounted on the robot's head, and the reflection from the floor is blocked by the robot. In addition, because the optical power reflected more than twice will be less than the noise power, it can be ignored [24]. In this study, only the primary reflection of the four walls of the room was considered, which can reduce the complexity of the light propagation path. This is simpler for VLP system design and implementation. Compared with multiple reflections, the transmission path stability of NLOS transmission is higher, and the signal quality and stability are relatively better. The received power $P_r$ of the PD during the transmission of the indoor LED light signal in the LOS link and NLOS link model could be expressed as [25] the following:

$$P_r = P_{LOS} + P_{NLOS}. \tag{20}$$

## 3. GRU Neural Network Model

As general recursion neural networks (RNNs) present the problems of long-term dependence and gradient explosion [26], Hochreiter and Schmidhuber proposed the long short-term memory (LSTM) neural network in 1997. This network contains input, forget, and output gates that control input, memory, and output values, respectively [27]. Therefore, the LSTM network can effectively solve the problem of gradient vanishing and gradient explosion and is highly effective for large-scale problem processing; thus, it is widely used. The GRU network was proposed by Kyunghyun Cho et al. in 2014. This is a highly effective variant of the LSTM network [28], and the basic GRU unit structure is shown in Figure 4.

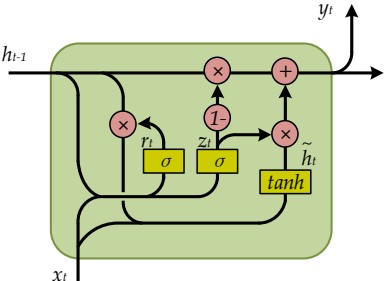

**Figure 4.** The basic unit structure of the GRU.

In a classical GRU network, the forward propagation equation at moment $t$ is as follows:

$$r_t = \sigma(x_t \cdot W_{rx} + h_{t-1} \cdot W_{rh} + b_r), \tag{21}$$

$$z_t = \sigma(x_t \cdot W_{zx} + h_{t-1} \cdot W_{zh} + b_z), \tag{22}$$

$$\widetilde{h_t} = \tanh(x_t \cdot W_{hx} + (r_t * h_{t-1}) \cdot W_{hh} + b_h), \tag{23}$$

$$h_t = (1 - z_t) * h_{t-1} + z_t * \widetilde{h_t}, \tag{24}$$

$$y_t = \sigma(W_o \cdot h_t + b_o), \tag{25}$$

where $\cdot$ and $*$ denote matrix multiplication and matrix dot product, respectively; $W_{rx}$, $W_{rh}$, $W_{zx}$, $W_{zh}$, $W_{hx}$, $W_{hh}$, and $W_o$ are the hidden layer weights; $b_r$, $b_z$, $b_h$, and $b_o$ are the hidden layer biases; $x_t$ is the input at moment $t$; $h_{t-1}$ is the hidden layer output state at moment $t-1$; $r_t$ and $z_t$ are the reset gate and update gate, respectively; $\widetilde{h_t}$ is the candidate set state at moment $t$; $h_t$ is the hidden layer output state at moment $t$; $y_t$ is the output at moment $t$; and $\sigma$ and tanh are activation functions. In general, $\sigma$ is a sigmoid function, which can be expressed as

$$\sigma(x) = \frac{1}{1 + e^{-x}}, \tag{26}$$

and tanh is a tangent function, which can be expressed as

$$\tanh(x) = \frac{e^x - e^{-x}}{e^x + e^{-x}}. \tag{27}$$

As with LSTM networks, GRU networks can also overcome the long-term dependency problem of traditional RNNs; however, the GRU network integrates the input and forget gates of the LSTM network into a single update gate, so the only two gates in the GRU network are the reset and update gates. In Equation (21), the reset gate $r_t$ controls the extent to which the hidden layer output state $h_{t-1}$ at moment $t-1$ is passed to the candidate set $\widetilde{h_t}$ at moment $t$. In Equation (22), the update gate $z_t$ determines the extent to which the output state $h_{t-1}$ at moment $t-1$ is carried to moment $t$. In Equation (23), the candidate set state $\widetilde{h_t}$ uses the reset gate $r_t$ to store past information. This is because the output of the reset gate will proceed through the sigmoid function, and each element in its output matrix is between 0 and 1, so the reset gate will control the size of the gate opening; a value closer to 1 indicates that more information is memorized. In Equation (24), the update gate $z_t$ determines how much of the candidate set state information $\widetilde{h_t}$ at moment $t$ and $h_{t-1}$ at moment $t-1$ will be retained, and the retained information is used as the output state information $h_t$ of the hidden layer at moment $t$. For Equation (25), using the hidden layer output state $h_t$ at moment $t$ as the output $y_t$ at moment $t$ is generally straightforward, i.e.,

$$y_t = h_t. \tag{28}$$

The output at time $t$ is passed to time $t+1$ to continue forward propagation as the input at time $t+1$.

We compared the commonly used recurrent neural networks, employing identical parameter settings. As shown in Table 1, ensuring prediction accuracy, the model complexity of the GRU network is lower than that of the LSTM model, which not only reduces the training parameters, but also accelerates the network training time.

**Table 1.** Comparison of positioning algorithms.

| Positioning Algorithm | Mean Squared Error | Average Error (m) | Maximum Error (m) | Training Parameters | Training Time (s) |
|---|---|---|---|---|---|
| SimpleRNN | 0.08891 | 1.02182 | 1.99923 | 5475 | 147.86 |
| GRU | 0.00038 | 0.02666 | 0.75596 | 16,923 | 172.91 |
| LSTM | 0.00045 | 0.03554 | 0.46776 | 21,675 | 234.57 |

## 4. Positing Process

### 4.1. Construction of Fingerprint Database

The robot moves in an indoor space area, and the maximum height during its activities is uncertain. In this study, we took the average height of a person, 1.7 m, as the maximum height during robot activity. Therefore, a volume of 4 m × 4 m × 1.7 m in the room was used as the positioning space, divided into sections of 0.18 m × 0.18 m × 0.18 m. The four vertices of each small square area after division were used as reference points, the robot head receiver model was placed at each reference point, and the top center point coincided with the reference point. We used three PDs to acquire optical signals and then filtered them. Thus, we obtained two signals of different frequencies and calculated their optical power values. Finally, we recorded the optical power value and position coordinates obtained at the reference point in the fingerprint database. The fingerprint data at the $k$-th reference point can be expressed as:

$$F_k = \begin{bmatrix} P_{k11} & P_{k12} & P_{k21} & P_{k22} & P_{k31} & P_{k32} & x_k & y_k & z_k \end{bmatrix}, \tag{29}$$

where $P_{kij}(i = 1, 2, 3; j = 1, 2)$ is the optical power value of the $j$-th LED light source received by the $i$-th PD at the $k$-th reference point, and $(x_k, y_k, z_k)$ are the position coordinates at the $k$-th reference point. Therefore, the VLP fingerprint database $F_{db}$ could be constructed as

$$F_{db} = \begin{bmatrix} F_1 & F_2 & \cdots & F_N \end{bmatrix}^T, \tag{30}$$

where $N$ is the number of reference points.

After dividing the positioning space into 0.18 m × 0.18 m × 0.18 m sections, the data obtained at the reference point were used as the training set. In addition, the positioning space was divided into 0.24 m × 0.24 m × 0.24 m sections, and the data obtained at this reference point were used as the test set. The training set was used to train the network model and provide it with a predictive ability, and the test set was used to evaluate the performance of the trained network model.

### 4.2. Data Preprocessing

GRU neural networks are very sensitive to input data, so we needed to normalize the input data. This process involved mapping the input data onto the same dimension, so that data of different dimensions had equal importance in the network. This not only improved the speed of network convergence, but also eliminated the influence of dimensions on the final result. We normalized the input data using

$$x_{norm} = \frac{x - x_{\min}}{x_{\max} - x_{\min}}, \tag{31}$$

where $x$ is the input data for the training set, $x_{\min}$ is the minimum value of all input data in the training set, $x_{\max}$ is the maximum value of all input data in the training set, and $x_{norm}$ is the normalized input data.

In addition, the GRU network required three-dimensional tensor inputs, so the input data needed to be converted into three-dimensional tensors before they were fed into the network. The input of the network was the optical power data, so the power data needed

to be converted into three-dimensional tensors. The converted $k$-th power data could be represented as

$$I_k = \begin{bmatrix} \begin{bmatrix} P_{k11} & P_{k12} \end{bmatrix} & \begin{bmatrix} P_{k21} & P_{k22} \end{bmatrix} & \begin{bmatrix} P_{k31} & P_{k32} \end{bmatrix} \end{bmatrix}. \tag{32}$$

Then, the input data could be expressed as

$$I = \begin{bmatrix} I_1 & I_2 & \cdots & I_n \end{bmatrix}^T, \tag{33}$$

where $n$ is the number of input data, and the shape of input data is $(n, 3, 2)$.

### 4.3. Selection of Performance Indicators

We used the mean squared error (MSE) and root mean squared error (RMSE) to evaluate the performance of the GRU network and VLP models.

The loss and evaluation functions of the GRU network model used MSE, which could effectively represent the error between the predicted and actual output of the network. In the process of neural network training, the gradient obtained by the loss function was input into the optimizer for gradient descent, and then the network weight was updated by backpropagation. We repeatedly trained the network to continuously improve its predictive capabilities. Finally, the test set was substituted into the trained network model for evaluation, and the network performance was evaluated by MSE. The MSE was calculated as follows:

$$e_{MSE} = \frac{1}{N} \sum_{i=1}^{N} \left[ (\hat{x}_i - x_i)^2 + (\hat{y}_i - y_i)^2 + (\hat{z}_i - z_i)^2 \right], \tag{34}$$

where $N$ is the number of sample sets, $(x_i, y_i, z_i)$ are the true values of the $i$-th sample point of the sample set, and $(\hat{x}_i, \hat{y}_i, \hat{z}_i)$ are the predicted values of the $i$-th sample point of the sample set.

In the positioning process, the RMSE could better reflect the relationship between the predicted and true positions, so the RMSE was used to calculate the VLP error. The RMSE between the true and predicted coordinates of the $k$-th reference point could be expressed as

$$e_k = \sqrt{(\hat{x}_k - x_k)^2 + (\hat{y}_k - y_k)^2 + (\hat{z}_k - z_k)^2}, \tag{35}$$

where $(x_k, y_k, z_k)$ are the true coordinates of the $k$-th reference point in the test set, and $(\hat{x}_k, \hat{y}_k, \hat{z}_k)$ are the predicted coordinates of the $k$-th reference point in the test set. Therefore, the average positioning error was

$$\bar{e} = \frac{1}{N} \sum_{k=1}^{N} e_k. \tag{36}$$

### 4.4. Building the GRU Network Model

We used the Python 3.9 compiler for the experiments and Tensorflow 2.6 and the Keras 2.6 deep learning framework to build the GRU network models. When building a network model, its initial weights are random, and so the predictions of the trained model differ each time. Therefore, in order to achieve reproducible experimental results, we had to fix the random seed before building the network model. In addition, in the process of network model construction, one must manually configure the number of GRU network layers and the number of neurons in the network layer. Furthermore, before training the network, one must also set the hyperparameters, such as the learning rate, number of iterations, and batch size. These parameters affect the complexity and performance of a model, so they need to be set appropriately. Below, we present the comparison and analysis of different hyperparameter values.

To explore the influence of the number of neurons on the accuracy of the model, we compared the values at intervals of eight.

As shown in Figure 5, the average positioning error was lower when the number of neurons in the GRU network layer was 24. However, the complexity of the model also increased when the number of neurons exceeded 24, and the average positioning error did not change significantly with an increase in the number of neurons. Therefore, the number of neurons in the GRU layer of the network model was set to 24.

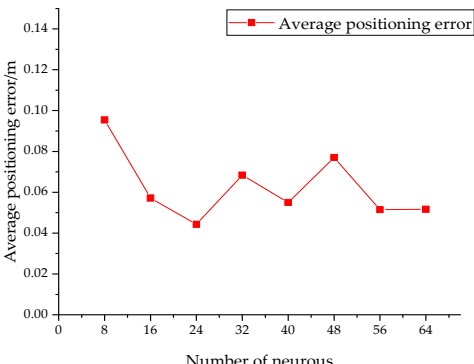

**Figure 5.** Average localization error for different numbers of neurons.

After settling on 24 network neurons, we analyzed the influence of the number of GRU network layers on the model performance.

From Table 2, one can see that the mean squared error and average localization error of the GRU network were smaller when the number of layers was two, and the model performance was improved. Furthermore, as the number of network layers increased, the error increased. When the number of layers is greater than two, increasing the number of layers of the network requires assigning more weights and training time to the network, which will lead to increased complexity of the network model and overfitting of the model, reducing the accuracy of the model. Therefore, we set the number of layers in the GRU network to two.

**Table 2.** The influence of the number of GRU network layers on the accuracy of the model.

| Number of Network Layers | Mean Squared Error | Average Error (m) |
| --- | --- | --- |
| 1 | 0.00483 | 0.11334 |
| 2 | 0.00082 | 0.04432 |
| 3 | 0.00203 | 0.08636 |
| 4 | 0.00231 | 0.07098 |
| 5 | 0.00467 | 0.14691 |

The batch size is the number of samples selected for training at one time, and back-propagation is performed by calculating the gradient of these samples, so it affects the degree of optimization and speed of a model.

In this study, the compared batch sizes were 16, 32, 64, 128, and 256. From Table 3, one can see that when the batch size was too small, the gradient of calculation was unstable due to the paucity of samples, and the network did not easily converge, causing the model accuracy to decrease. However, the network generalization ability was reduced when the batch size was too large, though the network model error did not change significantly. Table 3 also shows that the training time decreased as the batch size increased. According to our comparative analysis, the model was more effective when the batch size was set to 128.

**Table 3.** The influence of batch size on the accuracy of the model.

| Batch Size | Mean Squared Error | Average Error (m) | Training Time (s) |
|---|---|---|---|
| 16 | 0.00714 | 0.15795 | 2015.49 |
| 32 | 0.00143 | 0.08539 | 1036.73 |
| 64 | 0.00176 | 0.07599 | 617.59 |
| 128 | 0.00082 | 0.04432 | 388.47 |
| 256 | 0.00106 | 0.06665 | 247.98 |

Table 4 shows the effect of the learning rate on the model performance. The model performance was more favorable when the learning rate was set to 0.01, and the decreasing curve of the network loss function is shown in Figure 6.

**Table 4.** The influence of the learning rate on the accuracy of the model.

| Learning Rate | Mean Squared Error | Average Error (m) |
|---|---|---|
| 0.005 | 0.00091 | 0.04544 |
| 0.010 | 0.00082 | 0.04432 |
| 0.015 | 0.00151 | 0.07569 |
| 0.020 | 0.00193 | 0.08529 |
| 0.025 | 0.00724 | 0.18912 |

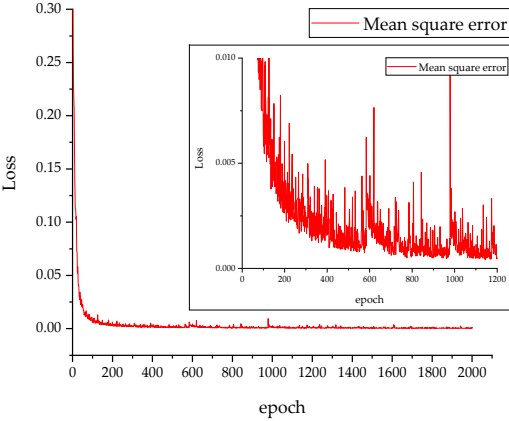

**Figure 6.** Loss function decline curve.

Figure 6 shows that when the number of iterations was around 950, the downward curve of the loss function was relatively flat, and there was no downward trend in subsequent iterations. To prevent overfitting and reduce training time, the maximum number of iterations of the network set to 950.

During network training, the gradient descent was slow when the learning rate was too small; thus, the training time needed to be increased to bring the model closer to the local optimum. However, the gradient decreased quickly when the learning rate was too large. Oscillation is easy in the later stage of training, but stabilization to local optimality is not straightforward, and gradient explosion may occur. In order to ensure that the network converged quickly at the beginning of training and more effectively at the end of training, we proposed a strategy to adjust the learning rate dynamically. Thus, the learning rate decay curve could be expressed as:

$$lr(epoch) = \frac{a}{1 + \exp(c(epoch - b))}, \tag{37}$$

where *epoch* is the iteration number of network training, and *a*, *b*, and *c* are set values, satisfying *a* > 0, *b* > 0, and *c* > 0. Here, *a* is the upper convergence boundary of the learning rate decay curve, and the value of $lr(0)$ is $a/(1 + \exp(-bc))$ when *epoch* = 0. If

$\exp(-bc) << 1$, $lr(0)$ is closer to $a$. Therefore, $a$ can be regarded as the initial learning rate. In this study, $a = 0.01$ was adopted. The value denoted as $b$ is the inflection point of the curve; $lr$ is larger in the interval of $epoch \in [0, b)$, so the gradient descent is faster and the network converges rapidly. Additionally, $lr$ decreases continuously after $epoch = b$, so the gradient descent slows down, which effectively suppresses the gradient oscillation it the late training period, and the network is more easily stabilized to the local optimum. The component $c$ is related to the decrease in the curve at the inflection point; the higher the value of $c$, the faster the curve falls at the inflection point. Based on continuous testing, the average positioning error was small when $a = 0.01$, $b = 700$, and $c = 0.02$, and the corresponding learning rate decay curve is shown in Figure 7.

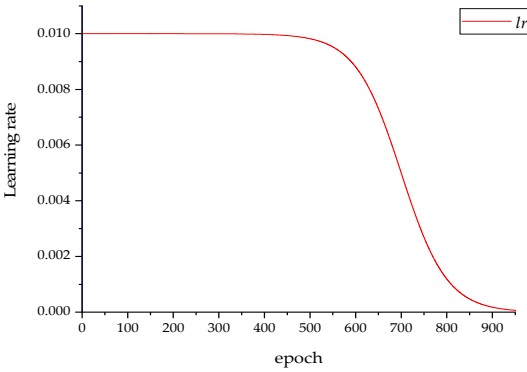

**Figure 7.** Learning rate decay curve.

As shown in Table 5, the learning rate decay strategy proposed in this paper corresponded to a higher VLP system accuracy, indicating that the method was effective.

**Table 5.** The effect of the proposed learning rate decay strategy and the learning rate setting of 0.1 on the accuracy of the model.

| Learning Rate | Mean Squared Error | Average Error (m) | Training Time (s) |
|:---:|:---:|:---:|:---:|
| 0.01 | 0.00075 | 0.04131 | 169.09 |
| $lr$ | 0.00038 | 0.02660 | 172.91 |

Therefore, the GRU network model was constructed according to the parameters established above, and its structure is shown in Figure 8.

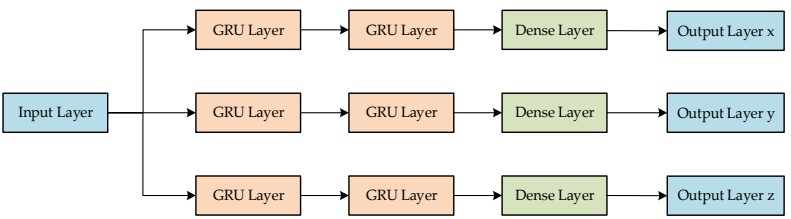

**Figure 8.** Structure of the GRU network model.

The model contained an input layer and three output layers, that is, the power data were input into the network, and the output comprised three coordinates. The hidden layer used three identical network structures, each containing two GRU network layers. In order to transform the data format of the GRU layer output into the final output data format, a dense layer was added before the output layer, and the network model parameters are shown in Table 6.

**Table 6.** GRU network model parameters.

| Parameter | Value |
|---|---|
| Number of neurons in the GRU layer | 24 |
| Number of neurons in the dense layer | 1 |
| Batch size | 128 |
| Number of iterations | 950 |
| Learning rate | Equation (37) |
| Optimizer | Adam |

## 5. Simulation Results and Analysis

To verify the localization performance of the proposed algorithm, a simulation environment was built according to the indoor visible-light localization model in Figure 1. We placed the hemispherical surface receiver model at each reference point in the positioning space and used three PDs on the hemispherical surface to acquire the signals sent by the two LEDs. The simulation parameters are shown in Table 7.

**Table 7.** Main parameters of simulation experiment.

| Parameter | Value |
|---|---|
| Room size (length $\times$ width $\times$ height) | 4 m $\times$ 4 m $\times$ 3 m |
| Height of positioning space | 0–1.7 m |
| (Training, testing) partition | (0.18, 0.24) m |
| LED position (x, y, z) | (1, 2, 3); (3, 2, 3) |
| LED semi $-$ angle at half $-$ power ($\phi_{1/2}$) | 30° |
| Amplitude of LED signal | 10 V |
| Frequency of LED signal | 4 KHz and 5 KHz |
| Effective area of PD ($A_{PD}$) | $10^{-4}$ m$^2$ |
| Azimuth angle of PDs ($\alpha_1, \alpha_2, \alpha_3$) | 0°, 135°, 225° |
| Radius of the robot receiver model ($r$) | 0.15 m |
| Arc length from PD to the top center point ($l$) | 0.05 m |
| Gain of optical filter $T_s(\psi)$ | 1 |
| Refractive index of optical concentrator ($n$) | 1.5 |
| FOV of PD ($\psi_{FOV}$) | 90° |
| Refractive index ($\rho$) | 0.8 |
| Reflection surface element area ($\Delta A$) | 0.0225 m$^2$ |
| Filter sampling frequency | 15 KHz |
| Type of filter | Butterworth bandpass filter |

In the simulation, the LED emitted a cosine AC signal, and to ensure that the LED communicated while achieving normal lighting, we added a DC bias to the LED signal. At the receiving end, the phase of the AC signal received by the PD was related to the transmission path of the signal, and the phase of the received signal differed each iteration. To be realistic, a phase shift of $kT$ was implemented for the LED emission signal in the simulation, where $k \in [0, 1)$ is a randomly generated value and $T$ is the LED emission signal period.

We obtained the simulated fingerprint data from the VLP model, and the sizes of the training and testing sets were 5290 and 2312, respectively. The training set was substituted into the GRU neural network to train the model, and after the training was completed, the testing set was substituted into the trained model to predict the position. The three-dimensional positioning predicted using the GRU network model for the LOS link and LOS + NLOS link scenarios is shown in Figure 9.

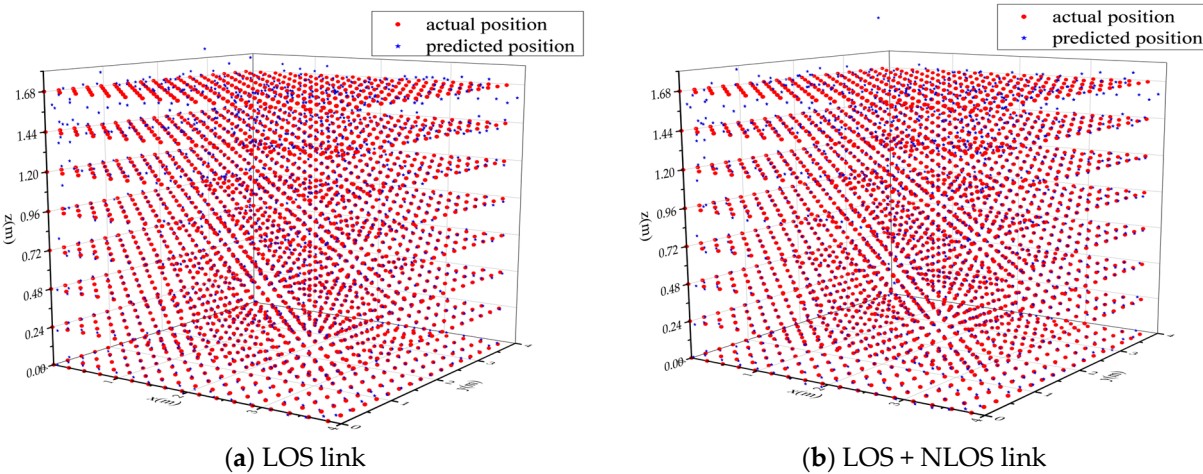

(**a**) LOS link         (**b**) LOS + NLOS link

**Figure 9.** Three-dimensional positioning predicted by GRU model.

Figure 9a,b show that as the positioning height increases, the deviation of the predicted location point from the actual location point increases. The positioning results in the corners are relatively poor. In addition, by comparing the positioning results in the z-axis direction of the LOS link and the LOS + NLOS link at a positioning height of 1.68 m, we find that the positioning results in the LOS + NLOS link are better.

Table 8 shows that the average localization error of the VLP model was 2.69 cm when only the LOS link case was considered, while the average localization error was 2.66 cm when both the LOS and NLOS link cases were considered. Figure 10 indicates that 95% of the positioning error was within 7.88 cm, showing that the model achieved centimeter-level positioning accuracy and met the needs of indoor positioning for robots.

**Table 8.** Performance comparison of 3D indoor visible-light localization models under different links.

| Link | Mean Squared Error | Average Error (m) |
|---|---|---|
| LOS | 0.00045 | 0.02687 |
| LOS + NLOS | 0.00038 | 0.02660 |

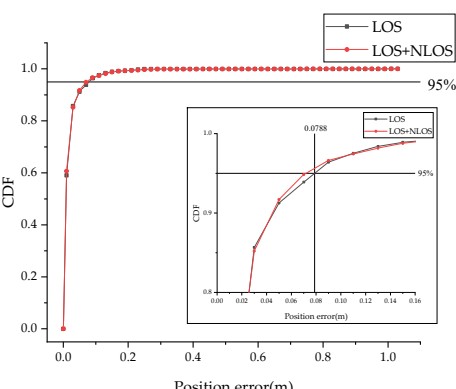

**Figure 10.** Cumulative distribution of positioning errors for LOS and LOS + NLOS links in 3D visible-light positioning system.

In the study, we used the same GRU network structure to make separate predictions for x, y, and z coordinates. To study the GRU network's prediction of x, y, and z coordinates, we analyze each coordinate error distribution separately. As can be seen from Figure 11, 90% of the errors in LOS + NLOS links are within 0.0265 m. Among them, the error in predicting the x-coordinate is the largest. As can be seen from Figure 1, the arrangement of

LEDs in the x-axis direction has a greater influence on the optical signal received by the receiver.

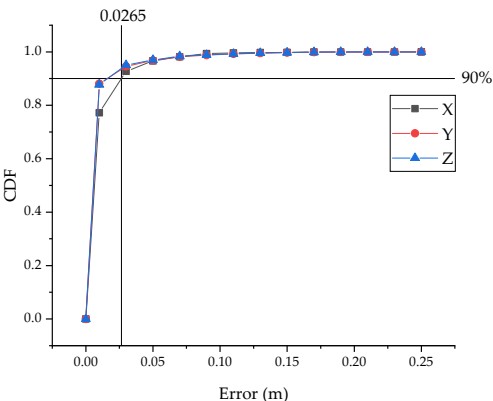

**Figure 11.** Cumulative distribution of errors for predicting three coordinates in a LOS + NLOS link.

To analyze the influence of the height on the accuracy of the model, we compared the two-dimensional positioning errors of the planes corresponding to different positioning heights. Table 9 shows the average and maximum positioning errors corresponding to the two-dimensional planes with the receiver placed at different heights under the LOS and LOS + NLOS link scenarios. When the positioning height was 0.24 m, the average positioning error of the model was the smallest for both LOS and LOS + NLOS links: the minimum values were 1.32 cm and 1.34 cm, respectively, and the maximum errors were 8.72 cm and 6.9 cm, respectively. However, when the positioning height was 1.68 m, the average positioning error of the model was the highest for both LOS and LOS + NLOS links, with maximum values of 7.75 cm and 7.84 cm, respectively, and maximum errors of 101.65 cm and 75.6 cm, respectively.

**Table 9.** Comparison of 2D positioning errors at different positioning heights for LOS and LOS + NLOS links.

| Height (m) | LOS | | LOS + NLOS | |
|---|---|---|---|---|
| | Average Error (m) | Maximum Error (m) | Average Error (m) | Maximum Error (m) |
| 0 | 0.01672 | 0.08093 | 0.01771 | 0.08095 |
| 0.24 | 0.01324 | 0.08719 | 0.01347 | 0.06899 |
| 0.48 | 0.01420 | 0.08867 | 0.01384 | 0.09652 |
| 0.72 | 0.01752 | 0.13633 | 0.01614 | 0.14298 |
| 0.96 | 0.01976 | 0.23178 | 0.01946 | 0.22707 |
| 1.20 | 0.02436 | 0.22531 | 0.02308 | 0.24067 |
| 1.44 | 0.03169 | 0.18135 | 0.03071 | 0.18333 |
| 1.68 | 0.07747 | 1.01654 | 0.07839 | 0.75597 |

Figure 12 shows that 80% of the positioning errors were within 9.87 cm for different positioning heights under the LOS link and LOS + NLOS link scenarios, and 80% of the positioning errors were within 3.44 cm for positioning heights below 1.44 m. Moreover, the CDF curve of the positioning error produced by the proposed algorithm for the LOS and LOS + NLOS link scenarios was small, which indicated that the algorithm had a good generalization ability and robustness for locating different links. Therefore, we will only discuss the positioning results for LOS + NLOS links.

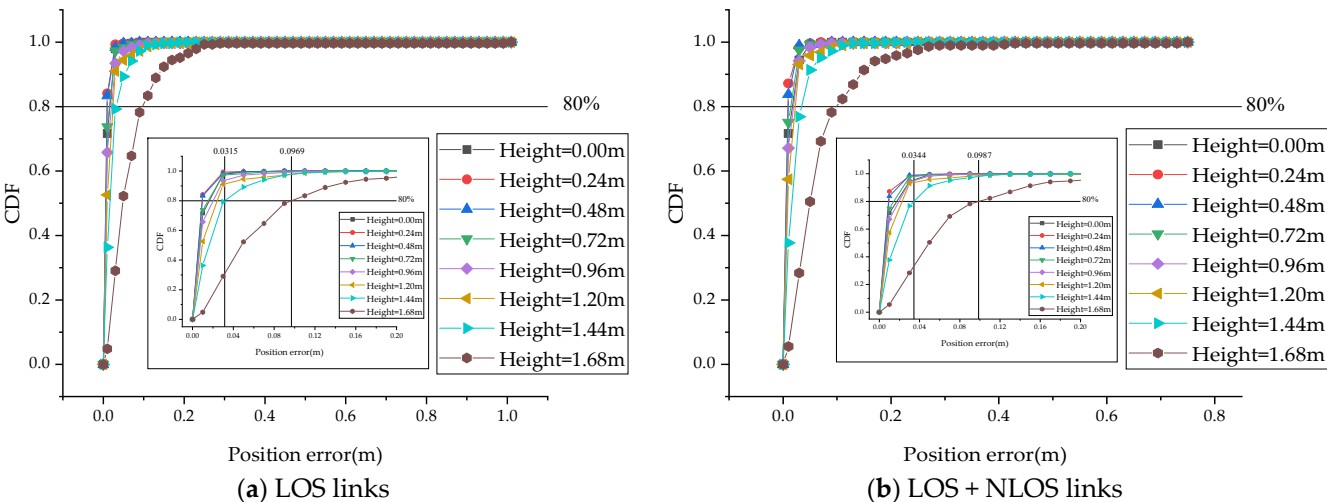

**Figure 12.** Cumulative distribution of two-dimensional positioning errors at different heights.

Figure 13 shows that when the positioning height was low, the errors were basically the same. When the positioning plane increased to a certain height, the positioning error also increased, and when the positioning height increased from 1.44 m to 1.68 m, this trend was more obvious. An analysis of Equations (15) and (27) reveals that the positioning error was mainly due to measurement errors related to the dc gain $H_{LOS}(0)$ and $H_{NLOS}(0)$ of the channel. When the positioning height increased, the emission angle of the LED light source also increased, and, according to Equations (4) and (12), this led to the higher attenuation of the optical signal, thereby increasing the error of the optical signal received by the PD and reducing the positioning accuracy.

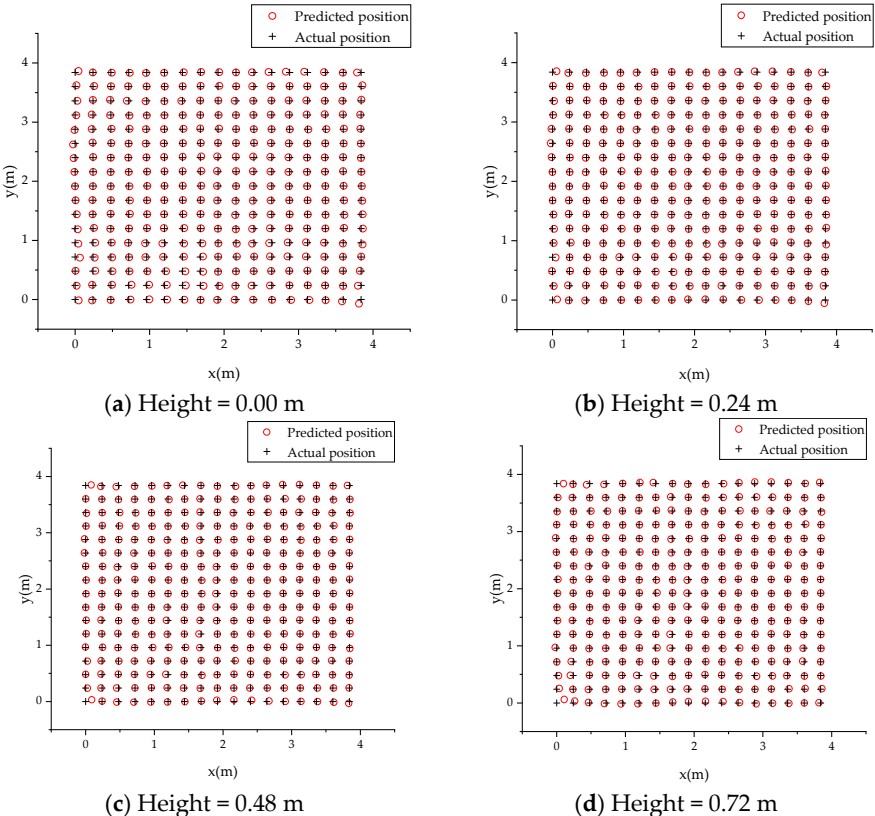

**Figure 13.** *Cont.*

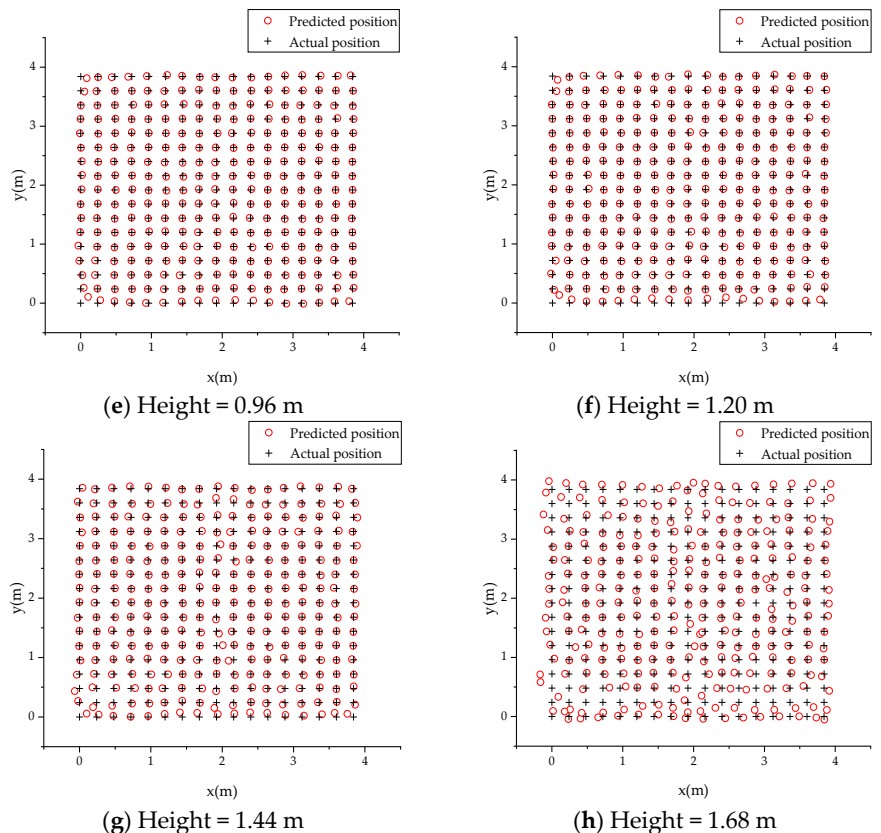

(**e**) Height = 0.96 m

(**f**) Height = 1.20 m

(**g**) Height = 1.44 m

(**h**) Height = 1.68 m

**Figure 13.** Comparison chart of 2D positioning results on different positioning heights under LOS + NLOS link.

## 6. Conclusions

We proposed an indoor visible-light three-dimensional positioning system based on a GRU neural network that solved the problem of the low positioning accuracy of existing robots. After the GRU network model was established, a learning rate attenuation strategy was proposed to improve the performance of the GRU network. A receiver placed on the robot's head was used to collect optical power data and then predict position coordinates from the trained GRU neural network. The experimental results showed that the average 3D positioning error was 2.69 cm when considering only LOS links, while the average error was 2.66 cm when considering LOS and NLOS links at the same time, and 95% of the positioning error was within 7.88 cm. For two-dimensional positioning with a fixed positioning height, 80% of the positioning error was within 9.87 cm. When the positioning height was 0.24 m, the average positioning error of the model under LOS and LOS + NLOS link scenarios was 1.32 cm and 1.34 cm, respectively. Therefore, the proposed method could achieve centimeter-level positioning accuracy to meet the needs of indoor robot positioning.

**Author Contributions:** L.Q.: conceptualization, investigation, supervision, resources, and writing (review). W.Y.: conceptualization, methodology, investigation, data curation, formal analysis, software, writing (original draft and editing), and validation. X.H. and D.Z.: resources and visualization. All authors have read and agreed to the published version of the manuscript.

**Funding:** This research was funded by the National Natural Science Fund Projects of China (62161041) and the Applied Technology Research and Development Fund Project of Inner Mongolia Autonomous Region (2021GG0104).

**Institutional Review Board Statement:** Not applicable.

**Informed Consent Statement:** Not applicable.

**Data Availability Statement:** Not applicable.

**Conflicts of Interest:** The authors declare no conflict of interest.

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
