# Peer review of "Indoor Visible-Light 3D Positioning System Based on GRU Neural Network"

_photonics, doi:10.3390/photonics10060633_

Round 1
Reviewer 1 Report
The authors investigated an indoor visible light three-dimensional positioning system based on gated recurrent units neural network. The accuracy of the proposed method needs to be improved. In addition, the following concerns should be addressed.
1. Is the proposed position estimation method unbiased or biased?
2. In Table 1, 2-layer neural network is shown to be with the best performance. However, the reason is missing.
3. The authors should compare the performance of the proposed method with those of other works.
Needs to be improved.
For example, the phrase "we believe" should not be shown in scientific papers.
Reviewer 2 Report
The authors present a solution to the existing robots' indoor positioning by proposing a visible light three-dimensional system based on the GRU neural network. The system uses a receiver attached to the robot's head to collect optical data, which is then utilized to predict position coordinates through a trained GRU neural network. A learning rate attenuation strategy is introduced to enhance the performance of the GRU network. The experimental results demonstrate that the proposed method can achieve centimeter-level accuracy.
Overall, I found the paper to be well-written and informative. However, I do have some comments and suggestions for improvement that I believe would strengthen the paper.
- While I appreciate the potential contribution of this work, I must note that there appears to be a lack of literature review related to the application of machine learning (ML) techniques in VLC systems. Are they different from the ML techniques applied in VLP?.
- It is unclear from the paper why the authors utilized a GRU network model to solve the VLP problem. There are many different types of machine learning techniques apart from GRU network models. What is the reasoning for using a GRU network?.
- I would recommend elaborating more on the primary reflective NLOS link description (lines 161-163).
- In lines 337-339, Figure 6 is described and the justification for setting the number of epochs to 950 is provided. However, it is unclear from the Figure whether the loss function remains constant at that value.
- Observing any difference between Figure 9 a) and b) is challenging, as no values representing the results are provided. It would be beneficial to highlight a specific area to gain a better understanding of the results.
- The caption for Figure 12 is misleading and requires an explanation as to why they represent cumulative distributions. Furthermore, I recommend replacing these figures with an alternative method of comparison for the positioning error as a function of height.
- As a reviewer, I would recommend that the authors include a sort of comparison with conventional VLP systems existing in the literature. This would provide valuable context for readers and allow them to better understand the contribution of the proposed VLP system.
- Also I noticed that there is a lack of discussion on the potential limitations of VLP based on machine learning. It would be helpful to discuss any factors that could affect the accuracy of the VLP system. Furthermore, it would be valuable to provide suggestions for how these limitations could be addressed in future research.
I believe that addressing these suggestions would greatly improve the quality and impact of your paper.
Reviewer 3 Report
The paper proposes an indoor visible light three-dimensional positioning system based on gated recurrent unites (GRU) neural networks to deal with nonlinearity in the system.
The topic is timely and interesting. However, the paper lacks some depth in the treatment. The complexity of the proposed approach should be investigated and the results should be compared to state-of-the-art three-dimensional approaches and not to results based on two-dimensional approach. In addition, the paper is not well-written. It contains many typos, grammatical mistakes, and long sentences that do not read well.
The English language must be improved. The manuscript is full of typos, grammatical mistakes, and long sentences that do not read well.
Reviewer 4 Report
To improve the accuracy of robot positioning, this paper proposes a 3D VLP system using GRU. My comments can be seen as below:
(1) More relative works particularly on NLOS VLP system with experimental results need to be introduced.
(2) It is not clear why to use the GRU. Why not using more simple mode such as CNN ?
(3) It is better to use experimental results instead of simulation results. It is not convincing.
Round 2
Reviewer 1 Report
In the response, the reason for biased estimation is incorrect. Please double check the definition of biased and unbiased estimations. It would be better that the authors derive a closed-form estimator; otherwise the estimation mean under considerable samples should be compared with the actual location to show whether the estimation is biased or not.
Minor editing of English language required
Reviewer 2 Report
Upon reviewing the latest version of the paper, I kindly request that the authors further revise the manuscript to incorporate, in some way, the justification provided for Point 2. I firmly believe that including this justification will greatly assist the reader.
In my opinion, the response to Point 7 lacks sufficient detail. It is important for the author to make a greater effort in providing a brief context of the existing traditional VLP. This contextual information is crucial for readers to grasp the significance and relevance of the authors' proposed advancements.
I have no further comments.
Reviewer 3 Report
The revised manuscript has been significantly improved. The authors are encouraged to proofread once more after all the edits.
Author Response
It is an honor to be recognized by you and thank you very much for your guidance.
Reviewer 4 Report
no further comments
Author Response
It is a great honor to have you recognize this work, which will be my motivation in my next work!